# REV-ERBα mediates complement expression and diurnal regulation of microglial synaptic phagocytosis

Percy Griffin[1], Patrick W Sheehan[1], Julie M Dimitry[1], Chun Guo[2], Michael F Kanan[1], Jiyeon Lee[1], Jinsong Zhang[2], Erik S Musiek[1,3]*

[1]Department of Neurology, Washington University School of Medicine in St. Louis, St. Louis, United States; [2]Department of Pharmacological and Physiological Science, Saint Louis University School of Medicine, St. Louis, United States; [3]Hope Center for Neurological Disorders, Washington University School of Medicine in St. Louis, St. Louis, United States

**Abstract** The circadian clock regulates various aspects of brain health including microglial and astrocyte activation. Here, we report that deletion of the master clock protein BMAL1 in mice robustly increases expression of complement genes, including *C4b* and *C3*, in the hippocampus. BMAL1 regulates expression of the transcriptional repressor REV-ERBα, and deletion of REV-ERBα causes increased expression of *C4b* transcript in neurons and astrocytes as well as C3 protein primarily in astrocytes. REV-ERBα deletion increased microglial phagocytosis of synapses and synapse loss in the CA3 region of the hippocampus. Finally, we observed diurnal variation in the degree of microglial synaptic phagocytosis which was antiphase to REV-ERBα expression. This daily variation in microglial synaptic phagocytosis was abrogated by global REV-ERBα deletion, which caused persistently elevated synaptic phagocytosis. This work uncovers the BMAL1-REV-ERBα axis as a regulator of complement expression and synaptic phagocytosis in the brain, linking circadian proteins to synaptic regulation.

*For correspondence:
musieke@wustl.edu

Competing interests: The authors declare that no competing interests exist.

## Introduction

The circadian clock orchestrates 24 hr rhythms in various cellular processes through transcriptional-translational feedback loops in most cells of the body (*Takahashi, 2017*). At the core of the clock's positive limb is the bHLH-PAS transcription factor BMAL1, which heterodimerizes with CLOCK or NPAS2 to drive the transcription of large sets of clock-controlled genes (*Mohawk et al., 2012*). BMAL1 transcriptional targets include the negative limb feedback regulator CRY and PER proteins, as well as REV-ERBα and β, nuclear receptors which can also inhibit the actions of the positive limb (*Preitner et al., 2002*; *Retnakaran et al., 1994*). Disruption of this circadian machinery is associated with various pathophysiological states including cancer, diabetes, and neurodegeneration (*Everett and Lazar, 2014*; *Musiek and Holtzman, 2016*; *Sulli et al., 2018*). Deletion of BMAL1 abrogates circadian clock function and leads to an ~80% decrease in REV-ERBα expression in the brain (*Musiek et al., 2013*). REV-ERBα functions as a transcriptional repressor in many tissues and has been implicated in regulation of metabolism and inflammation (*Everett and Lazar, 2014*). Previous work from our group shows that deletion of BMAL1 or its downstream target REV-ERBα causes neuroinflammation and impaired brain functional connectivity (*Griffin et al., 2019*; *Musiek et al., 2013*). Diminished BMAL1 and REV-ERBα expression have also been described in mouse models of Alzheimer's disease (AD)(*Lee et al., 2020*; *Stevanovic et al., 2017*). In AD, memory-associated, synapse-rich regions such as the hippocampus are affected early in the disease course (*Braak et al., 2006*). Synaptic loss also precedes neuronal death in neurodegeneration (*Selkoe, 2002*). Circadian

dysfunction is also a well-described symptom of AD and other neurodegenerative diseases (*Musiek and Holtzman, 2016*; *Videnovic et al., 2014*). Therefore, elucidating the how clock proteins regulate synaptic health is an important step in understanding the connection between circadian dysfunction and neurodegeneration.

A wealth of recent studies have emphasized the critical role of the complement system of the brain in regulating neuroinflammation and synaptic integrity. Synapses labeled with the opsonins C1q and C3 (*Stevens et al., 2007*) were first described to be pruned by microglia during development (*Schafer et al., 2012*). C4 protein, encoded by the mouse *C4b* gene, also contributes to synaptic pruning by microglia in vivo (*Sekar et al., 2016*; *Comer et al., 2020*). Complement-dependent microglial synaptic pruning has also been implicated in the pathogenesis of neurodegenerative and neuropsychiatric diseases (*Hong et al., 2016*; *Litvinchuk et al., 2018*; *Sekar et al., 2016*; *Werneburg et al., 2020*). Microglial activation is subject to circadian regulation (*Fonken et al., 2015*; *Hayashi et al., 2013*), and we have previously described that deletion of BMAL1 or REV-ERBα can induce microglial activation (*Griffin et al., 2019*; *Musiek et al., 2013*). Given the roles of the clock in neurodegeneration and microglial activation, we explored a potential role of the core clock in regulating synaptic health. Herein, we establish a link between the BMAL1-REV-ERBα axis, complement expression, and microglial synaptic pruning in the hippocampus.

## Results

### Disruption of the BMAL1-REV-ERBα axis induces complement upregulation in multiple brain cell types

While analyzing our previously published transcriptomic dataset from global *Bmal1* knockout (BMKO) hippocampal tissue, we observed a striking upregulation of several complement transcripts, in particular *C4b* and *C3* (*Griffin et al., 2019*), two genes which are critical for synaptic phagocytosis (*Figure 1A*). Other complement-related transcripts including *C1qc, C1qb, C1qa, and C1ra* were also increased in BMKO hippocampus (*Figure 1A*). Analysis of second published microarray dataset from our group which was derived from cerebral cortex samples from brain-specific *Bmal1* KO mice (*Nestin*-Cre;*Bmal1*$^{f/f}$) mice (*Lananna et al., 2018*) also revealed specific upregulation of *C4b*, but less so for other complement factors (*Figure 1A*). *C4b* was similarly increased in cerebral cortex samples from 4mo tamoxifen-inducible global BMAL1 KO mice (*CAG*-Cre$^{ERT2}$;*Bmal1*$^{f/f}$) in which *Bmal1* was deleted at 2mo, demonstrating that this is not a developmental phenomenon (*Figure 1—figure supplement 1*).

To determine the cell type(s) in which BMAL1 deletion induces complement gene expression, we examined cerebral cortex tissue from pan-neuronal-(*CamK2a*-iCre;*Bmal1*$^{fl/fl}$) (*Izumo et al., 2014*), astrocyte- (*Aldh1l1*-Cre$^{ERT2}$;*Bmal1*$^{fl/fl}$) (*Lananna et al., 2018*), and microglia-specific (*Cx3cr1*-Cre$^{ERT2}$; *Bmal1*$^{fl/fl}$) (*Parkhurst et al., 2013*) BMAL1 knockout mice. Both astrocyte- and microglia-specific *Bmal1* KO mice were treated with tamoxifen at 2mo and harvested 2 months later. Notably, *C4b* mRNA was strongly induced in the neuron-specific *Bmal1* KO mice, while *C3* was not (*Figure 1B*). *C4b* was also induced to a lesser degree in astrocyte-specific *Bmal1* KO mice, although *C3* was not (*Figure 1C*). Under basal conditions, neither *C4b* nor *C3* was induced in microglia specific *Bmal1* KO mice (*Figure 1D*). Deletion of *Bmal1* in primary *Bmal1*$^{fl/fl}$ neuron cultures via infection with an AAV8-Cre viral vector (versus AAV8-eGFP control) suppressed the BMAL1 transcriptional target *Nr1d1* (which encodes REV-ERBα) by 85% and induced *C4b* expression but caused no increase in *C3* (*Figure 1E*). *Bmal1* deletion also increased expression of *Fabp7* (*Figure 1E*), a known target of REV-ERBα-mediated transcriptional repression (*Schnell et al., 2014*). As REV-ERBα mRNA (*Nr1d1*) is also suppressed by ~80% in the brain following brain-specific *Bmal1* deletion (*Figure 1—figure supplement 1*), our data suggested that the upregulation of *C4b* gene expression observed with loss of BMAL1 could be mediated by transcriptional de-repression as consequence of downstream REV-ERBα loss. Accordingly, global deletion of REV-ERBα caused striking increases in *C4b, C3*, and other complement transcripts in the hippocampus as assessed by microarray analysis (*Figure 1A*) and confirmed in separate samples by qPCR (*Figure 1F*). Increased C3 protein expression was observed in activated astrocytes (*Figure 1G*, *Figure 1—figure supplement 2*) in the hippocampus of 5mo REV-ERBα KO (RKO) mice. There was less overlap of C3 expression and the microglial marker Iba1 (*Figure 1H*), although microglial C3 expression did increase in RKO brain (*Figure 1—figure*

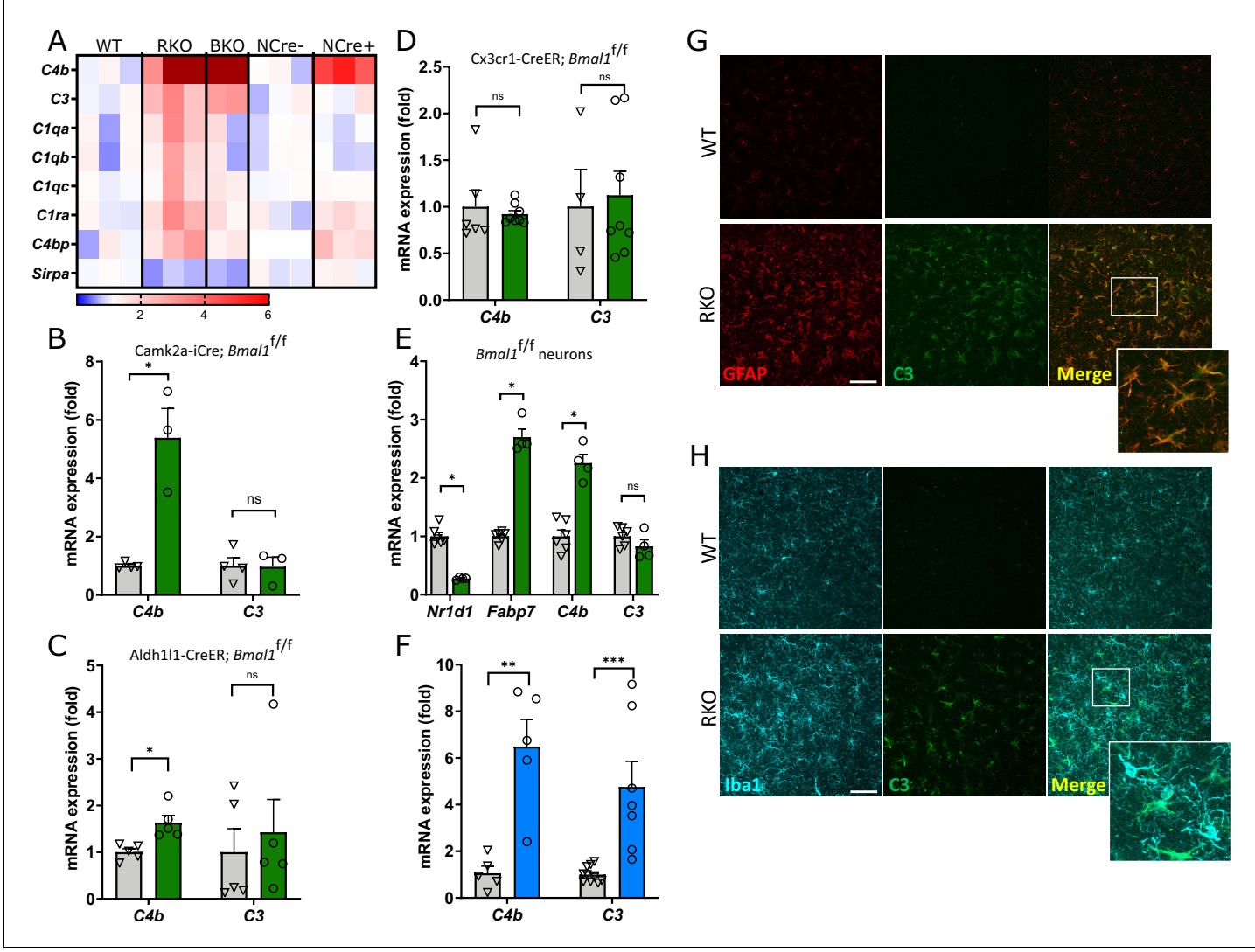

**Figure 1.** REV-ERBα regulates complement expression in multiple cell types downstream of BMAL. (**A**) Relative expression of complement-related transcripts taken from microarray analysis performed on hippocampus from 5mo REV-ERBα KO (RKO) and littermate WT mice (N = 3/genotype), global BMAL1 KO mice (BKO, N = 2), or 12mo *Nestin*-Cre;*Bmal1*[f/f] mice (NCre+), and littermate Cre- controls (NCre-, N = 3/group). Colors indicate fold change versus control. (**B**) qPCR analysis of 11 month old control (Cre-) and neuron-specific *Bmal1 KO* mice (*Camk2a*-iCre+;*Bmal1*[fl/fl]) for complement genes (N = 3–4/group). (**C**) qPCR analysis of control (Cre-) and astrocyte-specific *Bmal1 KO* mice (*Aldh1l1*-Cre[ERT2]+;*Bmal1*[fl/fl]) for complement genes (N = 5 mice/group). (**D**) qPCR analysis of control (Cre-) and microglia-specific *Bmal1 KO* mice (*Cx3cr1*- Cre[ERT2]+;*Bmal1*[fl/fl]) for complement genes (N = 4–8/group). For C and D, all mice (Cre- or +) were treated with tamoxifen at 2mo and harvested at 4mo, mixed sexes, Cre- littermates were used as controls. (**E**) qPCR analysis of mRNA from primary cortical neurons isolated from *Bmal1*[fl/fl] mice, treated with AAV8-GFP (control) or AAV8-Cre (n = 4–5 wells/group). (**F**) qPCR analysis of WT or RKO mouse hippocampal tissue for complement genes (N = 5–10 mice/group). (**G**) Representative 40X maximum intensity projections of GFAP and C3 staining as well as the merged channel in the hippocampus of 5mo WT or RKO mice and the associated normalized volumes for C3-GFAP staining (n = 8 mice, N = 4/group). (**H**) Representative 40X maximum intensity projections of Iba1 and C3 staining as well as the merged channel in the hippocampus of 5mo WT or RKO mice. Scale bar = 100 μm. *p<0.05 **p<0.01 ***p<0.001, ns = not significant, by two-tailed T-test with Welch's correction.

The online version of this article includes the following source data and figure supplement(s) for figure 1:

**Source data 1.** Data from the graphs depicted in *Figure 1*.
**Figure supplement 1.** Brain-specific or post-natal *Bmal1* deletion cause early induction of *C4b*.
**Figure supplement 2.** C3 colocalization with astrocytes and microglia in RKO brain.

supplement 2). The finding that *Bmal1* deletion in neurons (and to a less degree astrocytes) induces *C4b* mRNA but that *C3* only increases in global BMKO and RKO mice suggests that REV-ERBα directly represses *C4b* expression in neurons and astrocytes, but the induction of *C3* in both BMKO

and RKO brain is likely secondary to a multicellular inflammatory glial activation which occurs over time.

## REV-ERBα regulates microglial synaptic engulfment

We previously demonstrated that global REV-ERBα deletion induced microglial activation in vivo (*Griffin et al., 2019*). Given those results and the observation of increased *C4b* and *C3* expression in RKO brains, we examined the possibility that these changes would enhance synaptic phagocytosis in 4–6mo RKO mice. We primarily focused on the mossy fiber synapses in the CA2/3 region of the hippocampus, as these large synapses (thorny excrescences) can easily be stained and imaged using standard confocal microscopy. Triple-labeling of tissue sections was performed with antibodies against synaptophysin (a marker of presynaptic neuronal terminals – *Figure 2A*), CD68 (a microglial lysosome marker – *Figure 2B*) and Iba1 (to define microglial cell bodies and processes – *Figure 2C*). CD68 was used to ensure that the colocalized synaptic material was actually within the microglia phagosome. 3D reconstructions were made and total volumes of engulfed synaptic material were calculated (*Figure 2D*, *Figure 2—figure supplement 1*). In the RKO microglia, we observed synaptic material in the microglial process and cell body (*Figure 2Eii*, Eiii), whereas WT microglia only had engulfed synaptic material in the cell body (*Figure 2Ei*). In all, we noted a 13.6-fold increase in engulfed synaptic material in the hippocampus of RKO mice compared to their WT littermates (*Figure 2F*).

To corroborate these results, we also performed large area scanning electron microscopy (SEM) experiments of the CA3 region of 4–6mo WT and RKO mice. Using this method, we visually confirmed an increased number of presynaptic terminals within or in contact with microglia in the hippocampus of RKO mice compared to WT (*Figure 2G and H*, 2H(i-v), 2I). We also noted downregulation in the expression of the *Sirpa* gene in our RKO mice which codes for the protein SIRPα (*Figure 1A*). SIRPα was recently described as a surface receptor on microglia that serves as a receptor for a 'do-not-eat-me' signal (*Lehrman et al., 2018*). Notably, *Sirpa* was not downregulated in *Nestin*-Cre-*Bmal1* KO mice (*Figure 1A*), which have normal microglial BMAL1 and REV-ERBα expression, suggesting a possible cell-autonomous effect of REV-ERBα on microglial *Sirpa* expression. Taken together, our results suggest that REV-ERBα deletion induces synaptic engulfment.

## CA3 synapses are reduced by REV-ERBα deletion

Following our observations of increased synaptic phagocytosis in the RKO mice, we investigated the status of the synapses in the CA3 region of the hippocampus. Synapses were double labeled by using the presynaptic marker synaptophysin and the postsynaptic marker homer1, followed by confocal imaging. In RKO mice, we observed a significant decrease in the synaptic volume in the CA3 region of the hippocampus by synaptophysin staining (*Figure 3A*), homer1 staining (*Figure 3B*), and their colocalization (*Figure 3C*) compared to their WT littermates. To further confirm these results, we counted synapses in large area SEM images from the stratum lucidum of WT or RKO mice. Again, we noted a decrease in the number of synapses in the RKO mouse CA3 by SEM, as compared to their WT littermates (*Figure 3D*).

To ensure that our observations were not due purely to changes in synaptophysin protein expression in the presynaptic terminal, synapses were also stained with a second presynaptic marker, synaptoporin, and quantified. Synaptoporin was used because it is enriched in the mossy fiber synapses of the hippocampus (*Singec et al., 2002*). Again, we observed a similar decrease in synaptic volume in CA3 of RKO mice compared to their WT littermates using synaptoporin staining (*Figure 3E*). Interestingly, we did not observe significant differences in synaptic volumes between WT and RKO mice in the CA1 region of the hippocampus (*Figure 2—figure supplement 1*), suggesting that some terminals may be more susceptible to loss than others. To determine whether the loss in synapses was due to a loss of neuronal cell bodies, we quantified the volume and width of neuronal nuclei of the dentate gyrus, which project to CA3, via NeuN staining. We found no significant difference between the neuronal nuclear width or volumes of the dentate gyrus between the WT and RKO mice (*Figure 3—figure supplement 2*). Moreover, we did not observe any change in 3-nitrotyrosine staining in the dentate gyrus, suggesting there is no increase in neuronal nitrosative stress in RKO mice (*Figure 3—figure supplement 3*). Taken together, our data suggest that deletion of REV-ERBα results in robust synaptic loss in the CA3 region without obvious neuronal loss.

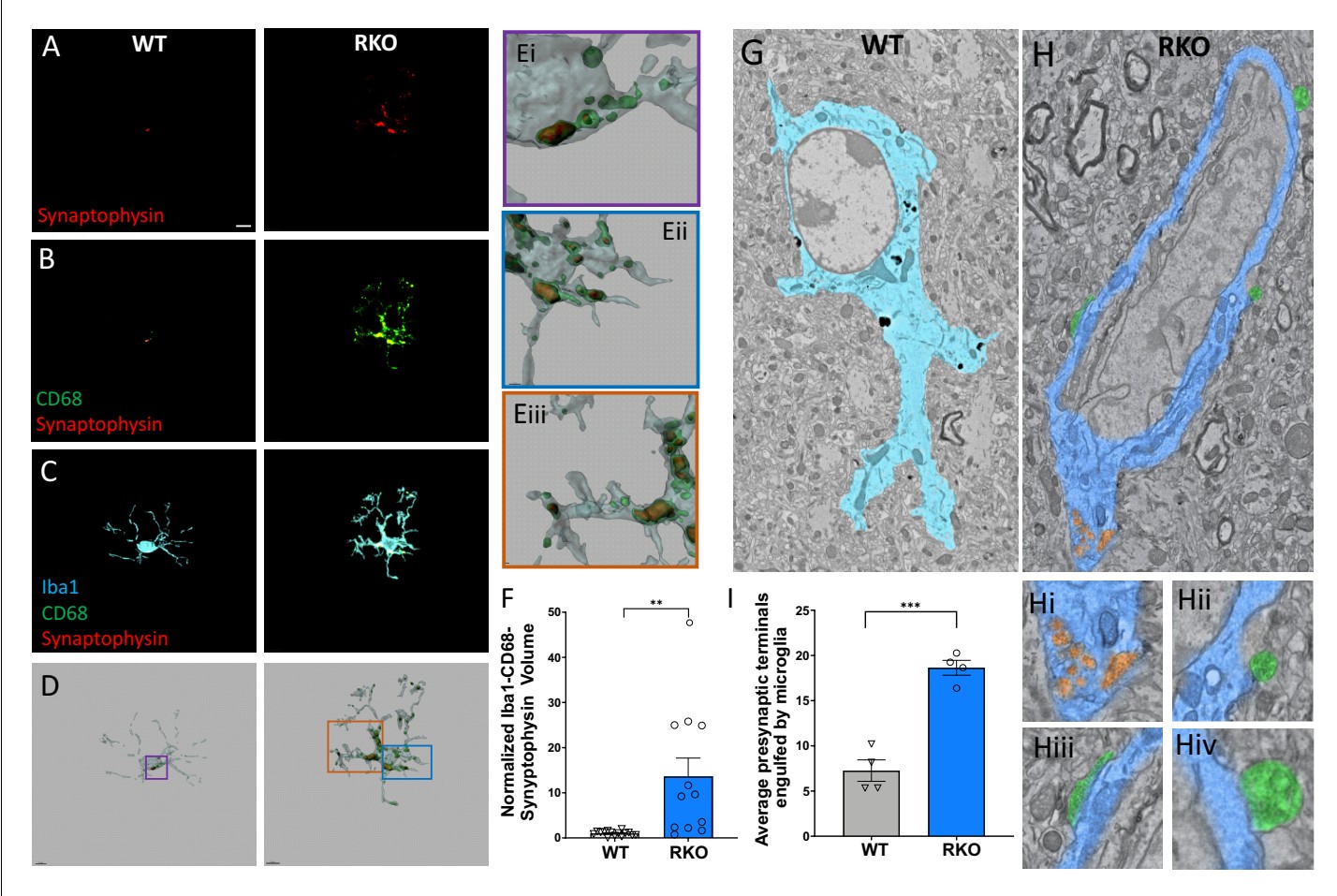

**Figure 2.** REV-ERBα deletion increases synaptic phagocytosis in the CA3 region of the hippocampus. (A-C) Image of synaptophysin (A), colocalized synaptophysin/CD68 (B), and colocalized synaptophysin/CD68/Iba1 (C) within microglia in the CA3 region of 4–6mo WT or RKO mice sacrificed at 11AM. D. Representative 3D surface rendering of microglia showing engulfed presynaptic material in lysosomes. (E) Magnified inset from WT (Ei) and RKO (Eii, iiii) microglia showing synaptophysin within CD68+ phagosomes. (F) Quantification of the normalized Iba1-CD68-Synaptophysin volumes from microglia in the CA3 region of the hippocampus of 4–6mo WT and RKO mice. Each point is the average of 3 sections from one mouse, N = 15 WT and 12 RKO mice. (G-H) Annotated, representative scanning electron micrographs of microglia in the CA3 of WT (G, pseudocolored Cyan) or RKO (H, pseudocolored Royal blue) mice sacrificed at 11AM with magnified inset pictures of engulfed presynaptic terminals (pseudocolored orange) in Hi and presynaptic terminals in contact with microglia (pseudocolored green) in Hii, Hiii, and Hiv. (I) Quantification of presynaptic terminals in contact with or engulfed by microglia in the CA3 of WT or RKO mice. Each point represents averaged data from a single mouse, N = 4 mice/genotype. In I, 3–15 microglia were counted per mouse. Scale bar for A–D = 5 μm. **p<0.01,***p<0.001 by two-tailed T-test with Welch's correction.

The online version of this article includes the following source data and figure supplement(s) for figure 2:

**Source data 1.** Data from the graphs depicted in *Figure 2*.
**Figure supplement 1.** Confirmation of antibody colocalization with orthogonal view.

## Time-of-day variation in microglial phagocytosis is regulated by REV-ERBα

In mouse cerebral cortex and ventral midbrain, REV-ERBα displays daily oscillation in mRNA level with its peak around zeitgeber time (ZT) eight and its trough around ZT20 (*Chung et al., 2014*). We have also previously described oscillations in microglial activation, with low activation at ZT5 and high activation at ZT17 (*Griffin et al., 2019*). Consequently, we investigated microglial synaptic engulfment at these times of day. WT (C57BL/6) and RKO mice were sacrificed ZT17 (11PM), and hippocampal sections were triple labeled with Iba1, CD68, and synaptophysin, and compared to those sacrificed at ZT5 (11AM, shown in *Figure 2*). Microglia from the CA3 region of the hippocampus in WT mice sacrificed at ZT17 showed significantly more engulfed presynaptic protein than those

at ZT5 (*Figure 4A–C*). When compared to the WT mice, RKO mice harvested at ZT5 and 17 showed a persistently increased level of synaptic engulfment in microglia, but with no time-of-day variation (*Figure 4C*), suggesting that daily oscillations in REV-ERBα may mediate rhythms in microglial synaptic phagocytosis. To further establish these findings, we used large area SEM to count the number of presynaptic terminals in contact with or within microglia at ZT17, and compared to mice harvested at ZT5 (shown in *Figure 2G–H*). In that experiment, we also noted a higher number of presynaptic terminals in contact with and within the microglia at ZT17 (*Figure 4D–F*). As a final confirmation of this finding, we injected *Cx3cr1*[GFP] mice, which express GFP in microglia (*Jung et al., 2000*), with an AAV-*Camk2a*-mCherry viral vector in the retrospenial cortex, to express mCherry in neurons. 4 weeks later, we collected brain samples at ZT5 and 17, and calculated the volume of mCherry+ material in individual GFP+ microglia. We observed that microglia at 11PM (ZT17) showed a qualitative increase in mCherry engulfment as compared to those at 11AM (ZT5) (*Figure 4—figure supplement 1*). Overall, our data establish a REV-ERBα-dependent rhythm for the engulfment of neuronal materials by microglia in the brain.

## Discussion

The current study shows that loss of the circadian protein BMAL1 causes upregulation of complement gene *C4b* in neurons and astrocytes, as well as increase astrocytic *C3* expression, changes which are recapitulated by loss of downstream REV-ERBα-mediated transcriptional repression. Deletion of REV-ERBα leads to microglial activation, increased *C4b* and *C3* mRNA expression, increased astrocyte C3 protein expression, and increased microglial synaptic phagocytosis in CA3 region of the hippocampus. Finally, we demonstrate a time-of-day variation in synaptic phagocytosis in the hippocampus of WT mice which is antiphase to REV-ERBα levels and lost after REV-ERBα deletion. Our findings suggest that the BMAL1-REV-ERBα axis regulates daily rhythms in synaptic phagocytosis, and that loss of REV-ERBα de-represses complement gene expression and locks the brain in a pro-phagocytic state.

The mechanisms by which BMAL1 deletion leads to increased *C4b* expression are complex and multicellular but likely depend on REV-ERBα, although contributions of other pathways cannot be excluded. REV-ERBα expression is decreased by ~85% following BMAL1 deletion, and REV-ERBα deletion phenocopies the complement gene expression increases seen in *Bmal1* KO brain. Indeed, it is well established that REV-ERBα functions as a transcriptional repressor (*Harding and Lazar, 1995*). Loss of REV-ERBα-mediated repression is a general mechanism governing increased transcript expression following BMAL1 deletion - as evidenced by strong upregulation of the REV-ERBα repression target *Fabp7* following BMAL1 deletion (*Schnell et al., 2014*). Notably, we queried an existing REV-ERBα ChIP-seq database from brain tissue (*Zhang et al., 2015*), and did not find any peaks corresponding to *C4b*. However, REV-ERBα is known to regulate transcription not only by direct binding to RORE and RevDR2 sites but also by regulating enhancer function and eRNA expression to alter gene expression in trans (*Lam et al., 2013*), and by altering chromatin looping (*Kim et al., 2018*). Thus, it is likely that REV-ERBα regulates *C4b* expression in one of these alternative ways which is not apparent by typical ChIP-seq. Tissue-specific deletion of BMAL1 shows that this BMAL1-REV-ERBα axis controls *C4b* expression in neurons and astrocytes, but not in microglia. Recent single-nucleus RNAseq data suggests that *C4b* is prominently expressed in oligodendrocytes (*Zhou et al., 2020*). However, we have not yet evaluated the effect of BMAL1 or REV-ERBα deletion on *C4b* in this cell type. Presumably, *C4b* expression is low in neurons and astrocytes under normal conditions but is increased by BMAL1 or REV-ERBα deletion. Work from the Stevens and McCarroll labs identified *C4b* expression in neurons which then tagged synapses and facilitated synaptic pruning, and showed that deletion of *C4b* prevents accumulation of C3 protein at synapses, as C4 protein in upstream of C3 activation in the classical complement pathway (*Sekar et al., 2016*). Moreover, neuronal overexpression of mouse C4 can drive microglial synaptic phagocytosis and cause functional connectivity deficits in mice (*Comer et al., 2020*), two phenotypes observed by us in RKO mice.

*Bmal1* KO and RKO mice at 5–6mo also show an increase in *C3* transcript and C3 protein in astrocytes. However, tissue-specific *Bmal1* deletion in neurons or astrocytes causes pronounced *C4b* increases but no *C3* increases, suggesting that C3 expression in astrocytes and microglia is a secondary response to increased neuroinflammation in aged global *Bmal1* KO or RKO mouse brain. Global

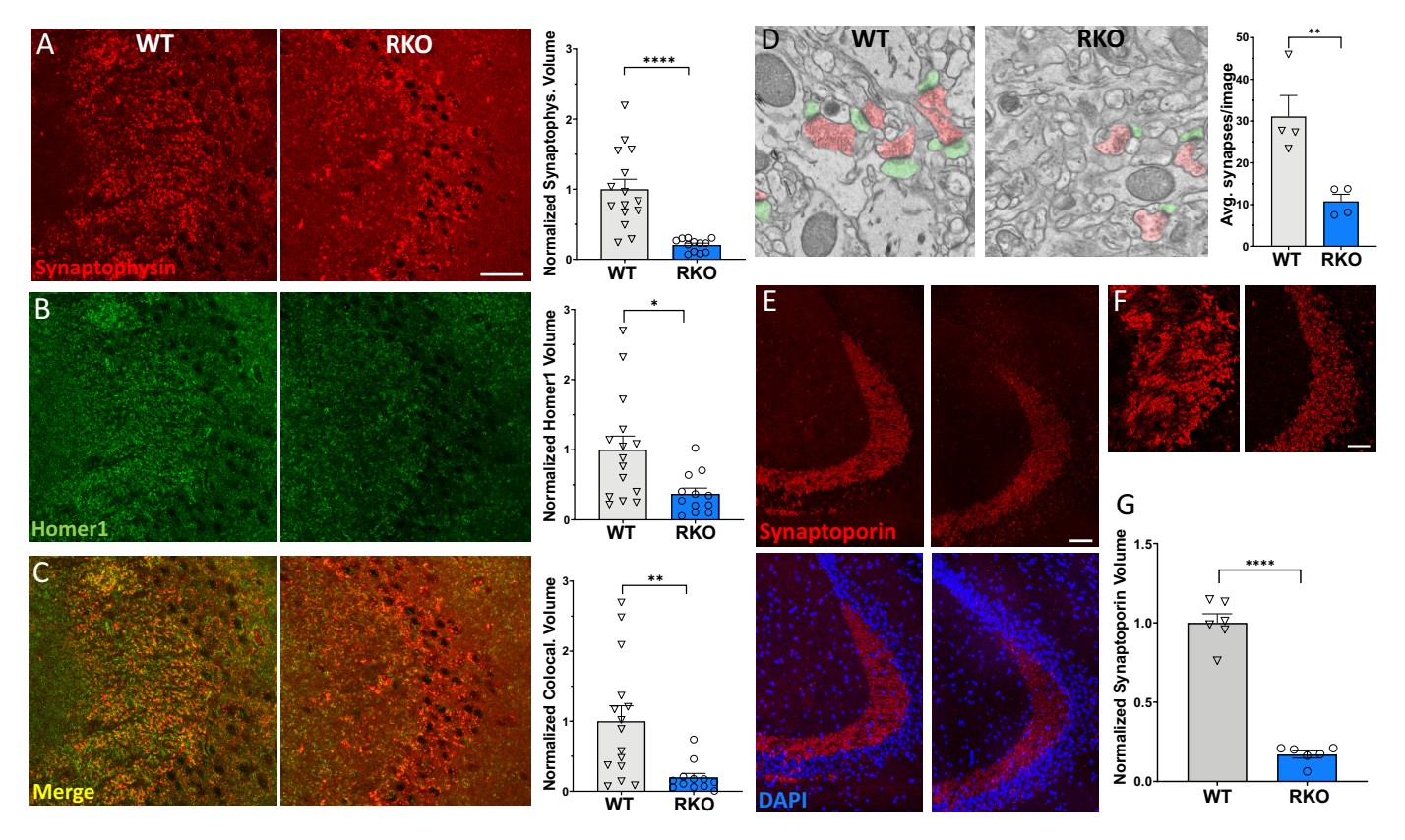

**Figure 3.** REV-ERBα deletion induces synapse loss in the CA3 region. (A) 60X representative maximum intensity projections showing synaptophysin staining in the CA3 of 4–6mo WT and RKO mice harvested at 11AM with the associated normalized volume quantification (N = 15 WT and 12 RKO mice). (B) 60X representative maximum intensity projections showing homer1 staining in CA3 of 4–6mo WT and RKO mice with the associated normalized volume quantification. (C) 60X representative maximum intensity projections showing colocalized synaptophysin and homer1 staining in CA3 of 4–6mo WT and RKO mice with the associated normalized volume quantification. (D) Representative scanning electron micrographs of synapses (presynaptic terminal pseudocolored in red, postsynaptic terminal in green), as well as the associated synapse counts for 4–6mo WT and RKO mice. N = 4 mice/genotype, with each point representing the average of 32–56 images counted per mouse. (E) 60X representative maximum intensity projections showing synaptoporin staining in the CA3 of 4–6mo WT and RKO mice. Colocalization with nuclei (DAPI) shown in lower images. (F) High-magnification images of synaptoporin staining of CA3 synapses from WT and RKO mice to show detail. (G). Normalized volume quantification of CA3 synaptoporin signal (N = 6 mice/genotype). *p<0.05, **p<0.01, ****p<0.0001 by two-tailed T-test with Welch's correction. In all panels, each point represent the average of three sections from a single mouse. Scale bar = 50 µm in all panels except D.

The online version of this article includes the following source data and figure supplement(s) for figure 3:

**Source data 1.** Data from the graphs depicted in *Figure 3*.

**Figure supplement 1.** REV-ERBα deletion does not induce synapse loss in CA1.

**Figure supplement 2.** No change in dentate gyrus neuronal layer volume or width in RKO mice.

**Figure supplement 3.** No change in dentate gyrus neuronal 3-nitrotyrosine in RKO mice.

inducible *Bmal1* KO mice also do not show *C3* induction at 2mo post-tamoxifen (despite high *C4b* expression), as these mice have not developed a full neuroinflammatory response at that age. Our data suggest that full induction of C3 in BMKO and RKO brain requires a multicellular, time-dependent inflammatory response involving microglia, as *Nestin*-Cre-*Bmal1* KO mice, which have *Bmal1* deletion in neurons and astrocytes but not microglia, have increased *C4b* expression but normal *C3* and *Sirpa* levels. We have previously shown that REV-ERBα deletion increases microglial activation, *Traf2* expression, and increased NFκB signaling (*Griffin et al., 2019*). Thus, REV-ERBα loss in microglia may be necessary, but not sufficient, to induce astrocyte C3 expression. Indeed, activated microglia can readily induce astrocyte C3 in response to LPS (*Liddelow et al., 2017*). NFκB signaling has also been shown to increase C3 expression and release from astrocytes (*Lian et al., 2015*). Functionally, C4 loss is associated with less synaptic tagging with C3 and thus less synaptic pruning

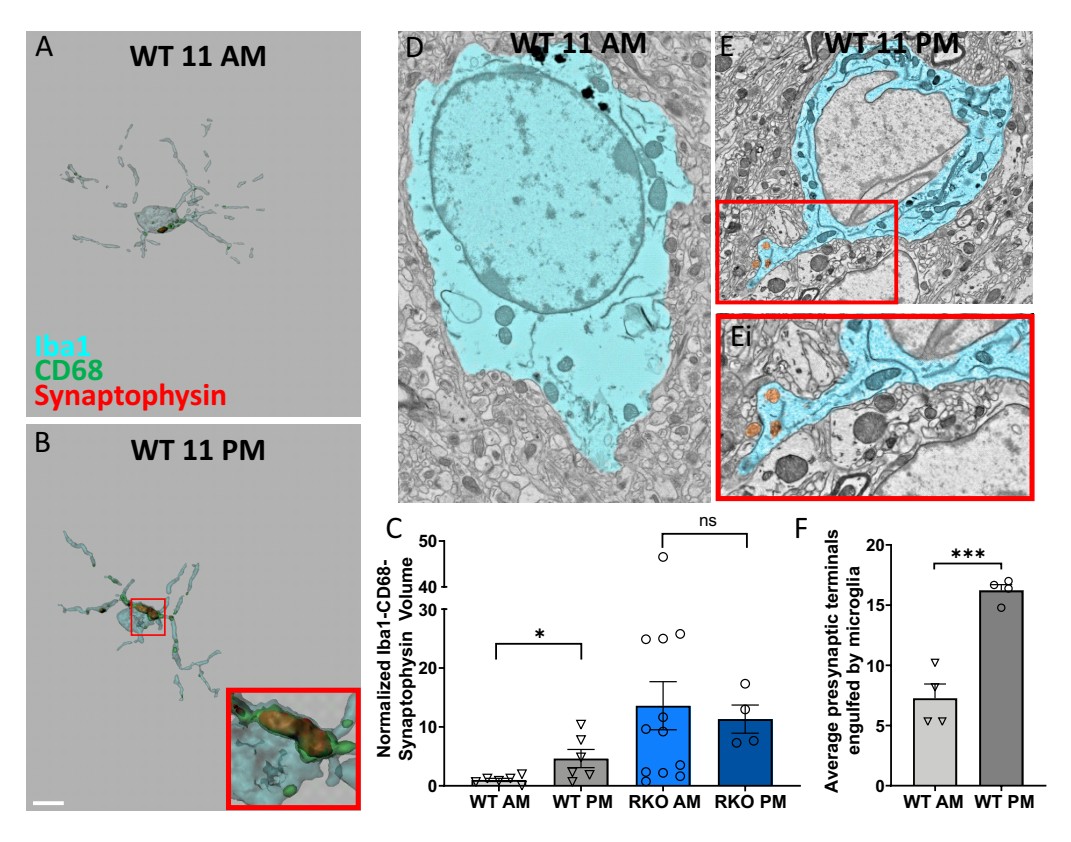

**Figure 4.** Time-of-day oscillation in microglial phagocytosis. Representative 3D surface rendering of Iba1, CD68 and synaptophysin in Iba1+ microglia from 6mo WT mice sacrificed at (**A**) 11AM or (**B**) 11PM with inset. Scale bar = 5 µm. (**C**) The normalized volume quantification of Iba1-CD68-Synaptophysin showing microglial phagocytosis of synapses in WT and RKO mice harvested at 11AM or 11PM (N = 4–12 mice/group). Note that the data from the WT AM and RKO AM groups was shown in *Figure 2F*. (**D-F**) Representative scanning electron micrographs of microglia in the CA3 of WT mice sacrificed at 11AM (**D**) or 11PM (**E**) as well as a high-magnification inset of presynaptic terminals within the microglia (**Ei**). (**F**) Quantification of presynaptic terminals in contact with or engulfed by microglia in WT mice sacrificed at 11AM or 11PM. N = 4 mice/timepoint, 4–10 fields of view each. Note that the WT AM group was shown in *Figure 2I*. *p<0.05, ***p<0.001 by two-tailed T-test with Welch's correction.

The online version of this article includes the following source data and figure supplement(s) for figure 4:

**Source data 1.** Data from the graphs depicted in *Figure 4*.
**Figure supplement 1.** Viral injection approach for diurnal variation in synaptic pruning.
**Figure supplement 2.** Diagram depicting multicellular contribution to synaptic phagocytosis by REV-ERBα.

(*Sekar et al., 2016*), suggesting that C4 expression can drive C3-dependent synapse pruning. Additionally, we noted that REV-ERBα regulated the microglial 'do-not-eat-me' signal *Sirpa* (encoding SIRPα) in our microarray data. Loss of inhibitory signaling from SIRPα makes microglia more likely to prune synapses, since SIRPα is primarily expressed on microglia in the CNS (*Lehrman et al., 2018*; *Zhang et al., 2014*). Therefore, our data suggest that de-repression of *C4b* in neurons and *Traf2* in astrocytes and microglia, as well as diminished *Sirpa* expression in microglia, lead to a 'perfect storm' of complement expression and microglial activation that promotes synaptic phagocytosis (*Figure 4—figure supplement 2*). Detailed future studies using combinations of tissue-specific REV-ERBα mutants will be needed to elucidate this complex interaction.

We focused on the CA2/3 mossy fiber synaptic boutons because they are large and easily labeled with synaptic vesicle markers – in this case synaptoporin and synaptophysin (*Grosse et al., 1998*). However, we did not observe synapse loss in the CA1 region, which may represent a regional variability in BMAL1-REV-ERBα mediated synaptic pruning. Certainly, it is possible that certain synapses are more susceptible to pruning than others, and this should be addressed in the future. We cannot exclude the possibility that REV-ERBα deletion causes neuronal damage, which elicits microglial phagocytosis of dysfunctional synapses. Indeed, neuronal *C4b* upregulation could be a damage

signal, although this has not been established. However, the absence of neuronal cell body loss or increased 3-nitrotyrosine in the dentate gyrus of RKO mice shows that there is no overt neurodegenerative response. These dentate gyrus granule cells give rise to the mossy fiber boutons in the CA3 (*Scharfman and Myers, 2012*), which are clearly decreased in RKO mice. Thus, the effect of REV-ERBα appears to be specific to the synapses.

Herein, we establish REV-ERBα as a regulator of microglial synaptic phagocytosis. However, we have previously reported that time-of-day changes in microglial morphology were abrogated by REV-ERBα deletion (*Griffin et al., 2019*). Herein, we observed that microglia engulfed more CA3 synapses at 11PM (ZT17) than at 11AM (ZT5). This was evident by both immunofluorescence and by electron microscopy. REV-ERBα deletion abrogated this time-of-day variation in synaptic protein phagocytosis, as RKO mice had high levels of phagocytosis at both timepoints. This data parallels our previous findings with microglial morphologic changes and suggests that REV-ERBα mediates daily changes in the degree of microglial synaptic engulfment, with both microglial activation and synaptic phagocytosis increasing when REV-ERBα levels are low. Accordingly, REV-ERBα deletion appears to lock the system in one of the extremes of this naturally occurring diurnal variation.

Several studies have illustrated the complexity of the interplay between cellular rhythms and sleep in regulating synapses in the brain (*Bellesi et al., 2017*; *Brüning et al., 2019*; *Noya et al., 2019*). RKO have subtle alterations in sleep but show grossly normal circadian rhythms in activity and are not sleep deprived (*Mang et al., 2016*), suggesting that sleep alteration is a possible but less likely contributor to the synaptic phenotypes in RKO mice. A previous study in the rat prefrontal cortex found that synaptic elimination was highest at ZT0 (*Choudhury et al., 2020*). While we did not examine ZT0, it is worth noting that these studies were done in the prefrontal cortex suggesting potential brain region-specific pruning patterns by time of day. Heterogeneity in synaptic pruning has already been described, with areas such as the cerebellum exhibiting greater microglial phagocytic capacity (*Ayata et al., 2018*). Future work will have to explore the patterns in microglial phagocytosis across more brain regions.

This work has clear implications for neurodegenerative and neuropsychiatric diseases, which have been linked to circadian disruption as well as complement dysregulation and synapse loss. Since REV-ERBα is a nuclear receptor with available small-molecule ligands (*Solt et al., 2012*), our findings suggest that it could be a therapeutic target for neurological and psychiatric disease. In previous studies in the brain, activation of REV-ERBs appears to suppress microglial cytokine production (*Griffin et al., 2019*) while inhibition of REV-ERBs can induce microglial amyloid-beta uptake and decrease plaque burden in mice (*Lee et al., 2020*). Effects on synaptic engulfment should be considered as REV-ERB-based therapeutics are developed.

## Materials and methods

### Key resources table

| Reagent type (species) or resource | Designation | Source or reference | Identifiers | Additional information |
|---|---|---|---|---|
| Antibody | Gfap (Rabbit polyclonal) | Dako/Agilent | Cat# Z0334 RRID:AB_10013382 | IF (1:2500) |
| Antibody | C3 (Rat monoclonal) | Novus | Cat# NB200-540 RRID:AB_10003444 | IF (1:500) |
| Antibody | Iba1 (Goat polyclonal) | Abcam | Cat# ab5076 RRID:AB_2224402 | IF (1:500) |
| Antibody | Synaptophysin (Mouse monoclonal) | Abcam | Cat# 8049 RRID:AB_2198854 | IF (1:100) |
| Antibody | Homer1 (Rabbit polyclonal) | Synaptic Systems | Cat# 160 003 RRID:AB_887730 | IF (1:500) |
| Antibody | CD68 (Rat monoclonal) | BioRad | Cat# MCA1957 RRID:AB_322219 | IF (1:250) |
| Antibody | NeuN (Mouse monoclonal) | EMD/Millipore | Cat# MAB377 RRID:AB_2298772 | IF (1:1000) |

*Continued on next page*

*Continued*

| Reagent type (species) or resource | Designation | Source or reference | Identifiers | Additional information |
|---|---|---|---|---|
| Antibody | 3-Nitrotyrosine (Rabbit polyclonal) | Millipore/ Sigma | Cat# AB5411 RRID:AB_177459 | IF (1:1000) |
| Genetic reagent (mouse) | *Bmal1 (Arntl)*-/- mice | Jackson Labs | Cat# 009100 - B6.129-Arntltm1Bra/J RRID:IMSR_JAX:009100 | |
| Genetic reagent (mouse) | *Bmal1*(flox/flox) mice | Jackson Labs | Cat# 007668-B6.129S4 (Cg)Arntltm1Weit/J RRID:IMSR_JAX:007668 | |
| Genetic reagent (mouse) | *Nr1d1*$^{-/-}$ mouse | Jackson Labs | Cat# 018447- B6.Cg-Nr1d1tm1Ven/LazJ RRID:IMSR_JAX:018447 | Rev-Erbα KO mouse line |
| Genetic reagent (mouse) | *Nestin*-Cre mice | Jackson Labs | Cat# 003771- B6.Cg-Tg(Nes-cre)1Kln/J RRID:IMSR_JAX:003771 | Nestin-Cre line crossed to Bmal1$^{f/f}$ |
| Genetic reagent (mouse) | *Aldh1l1*-Cre$^{ERT2}$ mice | Jackson Labs | Cat# 031008-B6N.FVB-Tg(Aldh1l1-cre/ERT2)1Khakh/J RRID:IMSR_JAX:031008 | Aldh1l1-CreER line crossed to Bmal1$^{f/f}$ |
| Genetic reagent (mouse) | *Cx3cr1*-Cre$^{ERT2}$ mice | Jackson Labs | Cat# 021160-B6.129P2(Cg)Cx3cr1tm2.1 (cre/ERT2)Litt/WganJ RRID:IMSR_JAX:021160 | Cx3cr1-CreER line crossed to Bmal1$^{f/f}$ |
| Genetic reagent (mouse) | *CAG*- Cre$^{ERT2}$ mice | Jackson Labs | Cat# 004682-B6. Cg-Tg(CAG-cre/Esr1*)5Amc/J RRID:IMSR_JAX: 004682 | CAG-CreER line crossed to Bmal1$^{f/f}$ |
| Genetic reagent (mouse) | *Camk2a*-iCre BAC TG mice | Obtained from Dr. J. Takahashi. Created by Dr. Gunther Schutz | MGI symbol: Tg(Camk2a-cre)2Gsc RRID:MGI:4457404 | BAC Tg Camk2a-iCre mouse |
| Sequence-based reagent | *Actb* (mouse) | Thermo-Fisher/Life Technologies | Taqman qPCR primer Cat#: Mm02619580_g1 | |
| Sequence-based reagent | *Nr1d1* (mouse) | Thermo-Fisher/Life Technologies | Taqman qPCR primer Cat#: Mm00520708_m1 | |
| Sequence-based reagent | *C4b* (mouse) | Thermo-Fisher/Life Technologies | Taqman qPCR primer Cat#: Mm00437893_g1 | |
| Sequence-based reagent | *C3* (mouse) | Thermo-Fisher/Life Technologies | Taqman qPCR primer Cat#: Mm01232779_m1 | |
| Sequence-based reagent | *Fabp7* (mouse) | Thermo-Fisher/Life Technologies | Taqman qPCR primer Cat#: Mm00437838_m1 | |
| Software, algorithm | Imaris | Bitplane, South Windsor, CT | RRID:SCR_007370 | Version 9, used for image analysis. |
| Software, algorithm | Prizm | GraphPad Software, LLC. | RRID:SCR_002798 | Version 8.3.0 |

## Mice

*Nr1d1*$^{+/-}$mice on C57bl/6 background were obtained from The Jackson Laboratory (Bar Harbor, ME) and bred at at Washington University. Heterozygous mice were bred together to generate *Nr1d1*$^{+/+}$ (wt) and *Nr1d1*$^{-/-}$ (referred to as REV-ERBα KO or RKO) littermates which were used for experiments. For all experiments, a mix of male and female mice was used. No obvious difference between sexes was noted, though experiments were not powered to detect sex differences. WT mice were C57Bl/6J mice from Jackson Labs. Constitutive *Bmal1*$^{-/-}$ (BMKO), CAG-Cre$^{ERT2}$, *Aldh1L1*-Cre$^{ERT2}$, *Cx3cr1*-Cre$^{ERT2}$, and *Bmal1*$^{fl/fl}$ mice were obtained from the Jackson Laboratory, and bred so that mice were heterozygous for Cre and homozygous for the floxed *Bmal1* allele. In these experiments, Cre-; *Bmal1*$^{f/f}$ littermates were used as controls. All inducible knockout lines were were treated with tamoxifen (Sigma) dissolved in corn oil via oral gavage, 2.5 mg/day for 5 days, at 2mo. *Camk2a-*

iCre+;*Bmal1*$^{fl/fl}$ mice were bred at University of Texas- Southwestern Medical Center. *Cx3cr1*$^{GFP}$ mice were obtained from The Jackson Laboratory. Mice were housed on a 12/12 light/dark cycle and fed ad libitum. All procedures performed on the mice were approved by the Washington University IACUC.

## Immunohistochemistry

The following primary antibodies were used (with dilution): Gfap (Rabbit polyclonal, 1:2500, Dako/Agilent Cat# Z0334), C3 (Rat monoclonal, 1:500, Novus Biologicals Cat# NB200-540), Iba1 (Goat polyclonal, 1:500, Abcam Cat# ab5076), Synaptophysin (Mouse monoclonal, 1:100, Abcam Cat# ab8049), CD68 (Rat monoclonal, 1:250, BioRad, Cat# MCA1957), Homer1 (Rabbit polyclonal, 1:500, Synaptic Systems Cat# 160 003), Synaptoporin (Rabbit polyclonal, 1:1000, Synaptic Systems Cat# 102 002), NeuN (Mouse monoclonal, 1:1000, EMD/Millipore Cat# 14-5698-82). Specificity of all antibodies was confirmed via staining without primary antibody, and in the case of the anti-C3 antibody, was tested in C3 KO mouse tissue.

   Mice were anesthetized with intraperitoneal (i.p.) injection of pentobarbital (150 mg/kg), followed by pump perfusion for three mins with ice cold Dulbecco's modified Phosphate Buffered Saline (DPBS) containing 3 g/l heparin sulfate. One hemisphere was drop fixed in 4% paraformaldehyde (PFA) for 24 hr at 4°C, then cryoprotected with 30% sucrose in PBS also at 4°C for at least 48 hr. Brains were embedded in OCT and frozen in acetone with dry ice. Twelve μm serial coronal sections were cut on a cryostat and mounted directly onto the glass slides for synaptic pruning analysis or synaptic quantification. Sections were washed three times in Tris buffered saline (TBS), blocked for 45 min in TBS containing 20% goat (or donkey) serum, 2% Mouse-on-mouse (M.O.M) blocking reagent and 0.4% Triton X-100 (Sigma-Aldrich, St. Louis, MO). Sections were then incubated in TBS containing 10% goat (or donkey) serum, 8% M.O.M protein concentrate and 0.4% Triton X-100 with primary antibody overnight at 4°C. Sections were then washed three times and incubated for 4 hr at room temperature with 1:1000 fluorescent secondary antibody in a solution of TBS containing 10% goat (or donkey) serum, 8% M.O.M protein concentrate and 0.4% Triton X-100. For other non-synaptic staining, 50 μm serial coronal sections were cut on a freezing sliding microtome and stored in cryoprotectant (30% ethylene glycol, 15% sucrose, 15% phosphate buffer in ddH$_2$O). Sections were washed three times in Tris buffered saline (TBS), blocked for 30 min in TBS containing 3% goat serum and 0.25% Triton X-100 (Sigma-Aldrich, St. Louis, MO) then incubated in TBS containing 1% goat serum and 0.25% Triton X-100 with primary antibody overnight at 4°C. Sections were then washed three times and incubated for 1 hr at room temperature with 1:1000 fluorescent secondary antibody and mounted on slides. Confocal images were taken on the Nikon Elements software on the Nikon A1Rsi scanning confocal microscope. Z-stacks were taken at a step size of 0.5–1 μm from dark to dark through the tissue.

## Microarray analysis

Previously-published transcriptomic datasets were analyzed for expression of complement transcripts. Both datasets were from Agilent 4 × 44 k mouse microarrays. One dataset was derived from 5- to 6-month-old Bmal1 and REV-ERBα KO mice with WT littermate controls, and used hippocampal tissue. This dataset was previously published (*Griffin et al., 2019*) and is freely available on the Array Express Server: https://www.ebi.ac.uk/arrayexpress/experiments/E-MTAB-7590. The second dataset was derived from *Nestin*-Cre;*Bmal1*$^{f/f}$ mice (and Cre- controls) aged to 12 months, and used cerebral cortex tissue. This dataset was also previously published (*Lananna et al., 2018*) and is also available on Array Express Server: https://www.ebi.ac.uk/arrayexpress/experiments/E-MTAB-7151. For this dataset, the samples harvested at 6pm are shown.

## Quantitative PCR

Flash-frozen brain tissue was homogenized with a mechanical handheld homogenizer for 20 seconds in RNA kit lysis buffer (PureLink RNA Mini Kit, Life Technologies, Carlsbad, CA) plus 1% β-mercaptoethanol. RNA was then purified using the kit protocol. Cells well collected and lysed in Trizol (Life Technologies). The aqueous layer was collected following chloroform extraction (added at 1:5 then spun at 13,000xg for 15 min) with RNA isolation protocol. RNA concentrations were then measured using the Nanodrop spectrophotometer and cDNA was made using a high-capacity RNA-cDNA

reverse transcription kit (Applied Biosystems/LifeTechnologies) with 1 µg RNA used per 20 µL reaction. Real-time quantitative PCR was performed with Taqman primers and PCR Master Mix buffer (Life Technologies) on the ABI StepOnePlus 12 k Real-Time PCR thermocyclers. β-actin (*Actb*) mRNA levels were used for normalization during analysis. The following primers were used (all from Life Technologies, assay number is listed): *Actb*: Mm02619580_g1, *Nr1d1*: Mm00520708_m1, *C4b*: Mm00437893_g1, *C3*: Mm01232779_m1, *Fabp7*: Mm00437838_m1.

## Electron microscopy

Animals were perfused with a fixative mix consisting of 2.5% glutaraldehyde + 2% paraformaldehyde (fresh, EM grade) in 0.15 M cacodylate buffer with 2 mM CaCl2 (final concentrations). Brains were extracted, blocked, and 100 µm coronal sections were made using a Leica 1200S vibratome. Tissues were then washed 3 × 10 min in cold cacodylate buffer containing 2 mM calcium chloride, and then incubated in a solution of 1% $OsO_4$ containing 3% potassium ferrocyanide in 0.3M cacodylate buffer with 4 mM calcium chloride for 1 hr in the dark. Following incubation, tissues were incubated for 20 min in a 1% thiocarbohydrazide (TCH) solution, rinsed 3 × 10 min in ddH2O at room temperature and thereafter placed in 2% osmium tetroxide (NOT osmium ferrocyanide) in ddH20 for 30 min, at room temperature. Tissues were washed 3 × 10 min at room temperature in ddH2O then placed in 1% uranyl acetate (aqueous) and incubated at 4° overnight. Tissues were incubated in a lead aspartate solution at 60°C oven for 30 min, washed 5 × 3 min in ddH2O, and returned to the lead aspartate solution at 60°C for 30 min. Tissues were washed 3 × 10 min in room temperature $ddH_2O$ and dehydrated in 50%, 70%, 90%, 100%, 100% acetone (anhydrous), 10 min each. Samples were embedded in Durcupan ACM resin and polymerized 60°C oven for 48 hr. Seventy nm slices were made using a Leica UC7 ultramicrotome, and sections were picked up on a Si wafer (Ted Pella, Redding, CA). Images were acquired on a Zeiss Merlin FE-SEM using a solid state backscatter detector (8kV, 900 pA) at 7 nm resolution with 5 µs pixel dwell times and 4x line averaging. Large area scans of ~150 µm x 150 µm field of view were acquired and stitched using Atlas 5.0 (Fibics, Ottawa, Canada).

## Stereotactic surgery and intracortical viral injections

Stereotactic viral vector injections were performed on *Cx3cr1*[GFP] mice. Briefly, mice were anesthetized with inhaled isoflurane, the head was shaved and the skull exposed and cleaned with iodine, and the head supported in ear bars. A hole was drilled through the skull. The coordinates used for the retrospenial cortex were 0.3 mm mediolateral (M/L), −2 mm anterior-posterior(A/P) and −1 mm dorso-ventral(D/V) with bregma as a reference. The needle was placed in the target location and then allowed to rest for 2 min before infusion. The AAV2-*Camk2a*-mCherry virus was obtained from the UNC viral vector core and infused at a rate of 0.2 µL/min. In total, the mice were injected with 2 µL in the retrospenial cortex. After the infusion, we waited 5 min for the virus to diffuse in the parenchyma and then the needle was slowly removed. The skin over the skull was then stitched up and antibiotic was applied to the area. The mice were then allowed to recover in an empty cage on a heating pad. After the surgery the mice were checked twice daily, 4 hr apart to ensure survival for 3 days. The mice were then allowed to age for 1 month before being sacrificed for sectioning and imaging.

## Synaptic volume and engulfment analysis/3D reconstructions

Imaris visualization and analysis software (Version 9, Bitplane, South Windsor, CT, USA) was used at the Washington University Center for Cellular Imaging. For all analyses, Z-stacks were saved in the. nd2 file format and loaded into the software. 3-D surfaces had a surface detail ranging from 0.1 to 0.3 µm. To quantify the volume of the synapses, we generated 3-D surfaces from each of the synaptic markers (Synaptophysin, Synaptoporin and Homer1). We used the Batch colocalization function to colocalize the Synaptophysin and Homer1 volumes. The total volume for each Z-stack was summed. For the synaptic engulfment analyses, we first generated volumes from synaptophysin, CD68 and Iba1 staining. We then colocalized the synaptophysin volume with the CD68 volume. The resulting volume was then colocalized with the Iba1 volume to produce a final microglial synaptic engulfment volume. All values were normalized to the average values for the control group in each experiment.

## Primary neuronal cultures

Neuronal cultures were made from age E18 *Bmal1*$^{f/f}$ mouse pups. Cortices plus hippocampi were dissected and stripped of meninges in ice-cold DMEM (Life Technologies) and then incubated in 0.05% Trypsin-EDTA at 37°C for 15 min. Tissue was gently triturated in 37°C DMEM plus 10% FBS (Gibco). Triturated cells were transferred to a second tube to remove debris, then diluted in Neurobasal (Life Technologies) plus B27 (Life Technologies) prior to plating on a PDL-coated plates. Cells were treated with 1 uM cytosine arabinoside for 24 hr, after which media was replaced. On DIV 4, Bmal1$^{f/f}$ neuron-enriched cultures were treated with either AAV8-CMV-GFP or AAV8-CMV-Cre for 4 days. Media was then again changed, and cells were harvested for in Trizol reagent at DIV 10.

## Statistical analysis

Statistical analyses were performed using GraphPad Prism 8. We performed t-tests with Welch's correction for multiple comparisons.

## Acknowledgements

This work was supported by awards from the Cure Alzheimer's Fund (ESM), Coins for Alzheimer's Research Trust (ESM), and NIH grants R01AG054517 and R01AG063743 (ESM). PG was supported by National Science Foundation Grant DGE-1745038. Imaging was performed with support from the Washington University Center for Cellular Imaging (WUCCI), which is funded by Washington University, Children's Discovery Institute (CDI-CORE-2015–505), and the Foundation for Barnes-Jewish Hospital (3770). The authors thank Drs. Mariko Izumo and Joseph Takahashi (UT Southwestern Medical School) for providing brain tissue samples from Camk2a-iCre;Bmal1(f/f) mice.

## Additional information

### Funding

| Funder | Grant reference number | Author |
| --- | --- | --- |
| National Institute on Aging | R01AG054517 | Erik S Musiek |
| National Institute on Aging | R01AG063743 | Erik S Musiek |
| Cure Alzheimer's Fund | Investigator award | Erik S Musiek |
| Coins for Alzheimer's Research Trust | Investigator award | Erik S Musiek |
| National Science Foundation | DGE- 1745038 | Percy Griffin |

The funders had no role in study design, data collection and interpretation, or the decision to submit the work for publication.

### Author contributions

Percy Griffin, Conceptualization, Data curation, Formal analysis, Investigation, Methodology, Writing - original draft, Writing - review and editing; Patrick W Sheehan, Formal analysis; Julie M Dimitry, Data curation, Investigation; Chun Guo, Investigation, Methodology; Michael F Kanan, Jiyeon Lee, Jinsong Zhang, Investigation; Erik S Musiek, Conceptualization, Supervision, Funding acquisition, Methodology, Writing - original draft, Project administration, Writing - review and editing

### Author ORCIDs

Erik S Musiek https://orcid.org/0000-0002-8873-0360

### Ethics

Animal experimentation: All experiments were conducted in accordance with recommendations in the Guide for the Care and Use of Laboratory Animals of the National Institutes of Health, and were approved by the institutional animal care and use committee (IACUC) at Washington University under protocol 2017-0124 (E. Musiek, PI).

Decision letter and Author response
Decision letter https://doi.org/10.7554/eLife.58765.sa1
Author response https://doi.org/10.7554/eLife.58765.sa2

## Additional files

### Supplementary files

• Transparent reporting form

### Data availability

The microarray data used in Fig. 1 is available on ArrayExpress E-MTAB-7590 and E-MTAB-7151. We have also uploaded all of the raw data from all of the figures in the paper as Source Files and to Dryad, which is available at: https://doi.org/10.5061/dryad.nzs7h44p1. This includes all of the image quantification data. Raw image files were not uploaded, as there are several hundred and they exceed 20GB.

The following dataset was generated:

| Author(s) | Year | Dataset title | Dataset URL | Database and Identifier |
|---|---|---|---|---|
| Musiek ES, Griffin P, Sheehan PW, Dimitry JM, Guo C, Kanan MF, Lee J, Zhang J | 2020 | REV-ERBα mediates complement expression and circadian regulation of microglial synaptic phagocytosis | https://doi.org/10.5061/dryad.nzs7h44p1 | Dryad Digital Repository, 10.5061/dryad.nzs7h44p1 |

The following previously published datasets were used:

| Author(s) | Year | Dataset title | Dataset URL | Database and Identifier |
|---|---|---|---|---|
| Musiek ES | 2014 | Microarray analysis of mouse cerebral cortex from brain-specific Bmal1 knockout and Per1;Per2 double mutant mice at 2 timepoints | https://www.ebi.ac.uk/arrayexpress/experiments/E-MTAB-7151/ | ArrayExpress, E-MTAB-7151 |
| Musiek ES | 2018 | Microarray of hippocampus from 5mo wt, Bmal1 KO and Nr1d1 (Rev-Erb-alpha) KO mice at a single timepoint | https://www.ebi.ac.uk/arrayexpress/experiments/E-MTAB-7590/ | ArrayExpress, E-MTAB-7590 |

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
