## [Decision Letter]

**Acceptance summary:**

While a connection between circadian regulation and immune system function is well appreciated, characterizing details of this interaction is an exciting step forward.

**Decision letter after peer review:**

Thank you for submitting your article "REV-ERBα mediates complement expression and circadian regulation of microglial synaptic phagocytosis" for consideration by *eLife*. Your article has been reviewed by two peer reviewers, and the evaluation has been overseen by Louis Ptacek as Reviewing Editor and Huda Zoghbi as the Senior Editor. The following individual involved in review of your submission has agreed to reveal their identity: Laura Fonken (Reviewer #1).

The reviewers have discussed the reviews with one another and the Reviewing Editor has drafted this decision to help you prepare a revised submission.

Summary:

Griffin et al. submit "REV-ERBα mediates complement expression and circadian regulation of microglial synaptic phagocytosis" for publication in *eLife*. Using global and targeted gene deletion of BMAL1 and REV-ERBα, the authors provide evidence for the role of REV-ERBα as an integral component in targeted synaptopathy of the CA3 region via indirect complement activation and suggest REV-ERBα's role as a repressor of complement activation. This work and further understanding of the BMAL1-REV-ERBα axis provide a new target for therapeutic intervention in neurodegenerative disease. While the data have pivotal implications and are important to the field, the paper lacks clarity in some of the claims made. We have the following suggestions for improvement:

Essential revisions:

1) One statistical issue that should be addressed throughout the paper is related to power of studies and analyses. For example:

a) In Figure 1A: n = 2 for BMAL1 KO mice. For these microarray data, and n=3 would be more suitable.

b) In Figures 2I, 3D, 4F, I, data should be analyzed on a per mouse basis with at least n = 3 mice. As is, n = 2 mice/genotype and statistics are performed on data points such as number of microglia or synapses (i.e. n=50 microglia counted/mouse, n=2 mice). These data should be analyzed as the average of those 50 microglia per mouse and compared to the average of other mice with an n=3 mice/genotype at minimum. All data should be displayed as an average of the sections analyzed from a single mouse. Representing the data any other way would be inappropriate.

2) Please note the sex of the mice. This should at minimum be mentioned in the Materials and methods (animals section). If females were included, it would be interesting to add sex as a covariate in analyses as there a sex differences in a number of neurodegenerative and neuropsychiatric disorders. If females were not included, this should be justified per NIH guidelines.

3) The figures would benefit from additional representative images. For example, in Figure 1, the separate channels should be shown in addition to the merged analysis. It was unclear from image 1G whether there was some bleed through of the channels. It would also be beneficial to show higher magnification images in 1G and H in order to better visualize the cell morphology. It was hard to see the microglia morphology with the Iba1 staining in 1H – although the microglia 3D reconstruction looked great in subsequent figures. Also, high levels of co-localization are reported in 1H but it was not apparent in the representative image. Perhaps add arrows?

4) In image 3E, it is hard to identify if these pictures are taken in the same region. It might be helpful to include a lower magnification image with a box identifying where you are zooming in to higher magnification.

5) "In the brain, REV-ERBα displays daily oscillations on the mRNA level – with its peak at zeitgeber time (ZT) 8 (2pm) and its trough at ZT20 (2am) (Chung et al., 2014)." – REV-ERBα, like other clock genes, likely displays distinct peak timing in different brain regions (for example, see PMID 26271538). Please adjust this sentence accordingly. Also, it is not necessary to include am and pm time in this sentence.

6) A few of the conclusions were overstated. For example, the first sentence of the Discussion states: "The current study shows that loss of the circadian protein BMAL1 causes upregulation of complement gene C4b in neurons and astrocytes, as well as increase astrocytic C3 expression, all via loss of downstream REV-ERBα mediated transcription repression." These experiments do not prove that BMAL1 causes upregulation of complement via REV-ERB (data is only correlative). In order to prove this occurs via REV-ERB, REV-ERB would need to be "rescued" in BMAL1 KO mice. This could potentially be tested with SR9009. Either this sentence should be adjusted or additional experiments performed. Also, the first sentence to the second paragraph should be changed (for the same reason).

7) The discussion on sleep, while interesting, seemed tangential to the results presented in this manuscript. Please remove.

---

## [Author Response]

Essential revisions:1) One statistical issue that should be addressed throughout the paper is related to power of studies and analyses. For example:a) In Figure 1A: n = 2 for BMAL1 KO mice. For these microarray data, and n=3 would be more suitable.b) In Figures 2I, 3D, 4F, I, data should be analyzed on a per mouse basis with at least n = 3 mice. As is, n = 2 mice/genotype and statistics are performed on data points such as number of microglia or synapses (i.e. n=50 microglia counted/mouse, n=2 mice). These data should be analyzed as the average of those 50 microglia per mouse and compared to the average of other mice with an n=3 mice/genotype at minimum. All data should be displayed as an average of the sections analyzed from a single mouse. Representing the data any other way would be inappropriate.

Regarding the array data in Figure 1A, we had originally included these two Bmal1 KO samples just for comparison to the Rev-Erbα KO samples, and we agree that at least 1 more is needed. Rather than rerun the array, we have now included data from a previous array on Bmal1 KO brain tissue that we have previously published (see revised Figure 1A). We examined the same complement transcripts in this other microarray dataset (published in Lananna et al., 2018), which is derived from brain-specific Bmal1 KO mice (Nestin-Cre;Bmal1(f/f) aged to 12mo). This second dataset also shows a strong upregulation of C4b, though not other complement components. Notably, Nestin-Cre;Bmal1(f/f) have Bmal1 expression preserved in microglia, suggesting that deletion of microglial Bmal1 may play a role in regulating some of the complement expression in the brain (such as C3). We have also altered the heatmap in Figure 1A to show each biological replicate, so it is clear that only 2 BMKO mice are present.

To address the issue of N vs. n in the EM experiments, we have repeated the experiments and added data from additional mice to bring the N to 4 for all experiments. We have altered the graphs such that the average from each mouse is shown, and statistics are now based on that.

For former Figure 4G-I, the time constraints needed to obtain more mice and virus, inject, and wait the appropriate time to repeat the experiment were prohibitive. Figure 4G-I is really just a third way of confirming the data that we present in Figure 4C and F. Thus, we have moved Figure 4G-I to the supplement, removed the statistical analysis, and clearly noted that it is a qualitative analysis meant as an adjunct to the data in Figure 4. If the reviewer is still concerned with this treatment, we can remove it entirely, though we feel that it still adds to the paper, even if the N is low.

2) Please note the sex of the mice. This should at minimum be mentioned in the Materials and methods (animals section). If females were included, it would be interesting to add sex as a covariate in analyses as there a sex differences in a number of neurodegenerative and neuropsychiatric disorders. If females were not included, this should be justified per NIH guidelines.

We used a mix of both male and female mice, and have now mentioned this in the Materials and methods. We attempted to mix them equally between experimental groups. We re-examined the data in Figure 2F by sex, as half of the mice were male, half female, and did not see a significant difference related to sex, though the experiment was generally not powered to detect sex differences.

3) The figures would benefit from additional representative images. For example, in Figure 1, the separate channels should be shown in addition to the merged analysis. It was unclear from image 1G whether there was some bleed through of the channels. It would also be beneficial to show higher magnification images in 1G and H in order to better visualize the cell morphology. It was hard to see the microglia morphology with the Iba1 staining in 1H – although the microglia 3D reconstruction looked great in subsequent figures. Also, high levels of co-localization are reported in 1H but it was not apparent in the representative image. Perhaps add arrows?

We appreciate this concern and have added additional images to make things more clear. We have now added each channel individually for Figure 1G and H, and have added a high-magnification inset image in both Figure 1G and H.

4) In Figure 3E, it is hard to identify if these pictures are taken in the same region. It might be helpful to include a lower magnification image with a box identifying where you are zooming in to higher magnification.

We agree that the representative images are a bit difficult to localize. In both cases, the images are taken of the CA3 mossy fiber region. We have now added lower-magnification images along with DAPI images to aid in localization (Figure 3E).

5) "In the brain, REV-ERBα displays daily oscillations on the mRNA level – with its peak at zeitgeber time (ZT) 8 (2pm) and its trough at ZT20 (2am) (Chung et al., 2014)." – REV-ERBα, like other clock genes, likely displays distinct peak timing in different brain regions (for example, see PMID 26271538). Please adjust this sentence accordingly. Also, it is not necessary to include am and pm time in this sentence.

Indeed, this is a good point, and we have not examined Rev-Erb phase across brain regions. We have previously shown Nr1d1 mRNA data from cerebral cortex, and Chung et al. showed Nr1d1 mRNA data from ventral midbrain, with similar peak/trough timing. Thus, we adjusted this sentence to read “In the mouse cerebral cortex and ventral midbrain, REV-ERBα displays daily oscillation on the mRNA level with its peak at zeitgeber time (ZT) 8 and its trough at ZT20”, and have removed the 2pm and 2am notes.

6) A few of the conclusions were overstated. For example, the first sentence of the Discussion states: "The current study shows that loss of the circadian protein BMAL1 causes upregulation of complement gene C4b in neurons and astrocytes, as well as increase astrocytic C3 expression, all via loss of downstream REV-ERBα mediated transcription repression." – These experiments do not prove that BMAL1 causes upregulation of complement via REV-ERB (data is only correlative). In order to prove this occurs via REV-ERB, REV-ERB would need to be "rescued" in BMAL1 KO mice. This could potentially be tested with SR9009. Either this sentence should be adjusted or additional experiments performed. Also, the first sentence to the second paragraph should be changed (for the same reason).

We appreciate this point, as a rescue experiment would be needed to definitively say that loss of Rev-Erbα mediates increases in C4b in Bmal1 KO mice. Bmal1 deletion results in functional near-deletion of Rev-Erbα (~80% decrease in mRNA, see Figure 1—figure supplement 2), so it is likely that the increase in *C4b* (which occurs in both BMKO and RKO brains) is due to loss of Rev-Erb. However, some Rev-Erb-independent effect may contribute in BMKO mice. We have adjusted the statements to reflect the realities of the data: “The mechanisms by which Bmal1 deletion leads to increased complement gene expression are complex and multicellular and likely depend on REV-ERBα, though contributions of other pathways cannot be excluded.”

Our concern with overexpression of Rev-Erbα in the BMKO brain is that titrating the amount of Rev-Erb expression such that it returns to basal levels (as opposed to gross overexpression) would be very difficult, and supra-physiological overexpression of Rev-Erb will suppress C4b expression artificially without necessarily excluding other pathways activated by Bmal1 KO. However, we can undertake this experiment if the reviewers think it is critical to the paper. Unfortunately, because Bmal1 deletion causes ~80% loss of Rev-Erbα/β expression, SR9009 generally does not work in general Bmal1 KO mice in our hands (as there is very little REV-ERB protein to activate).

7) The discussion on sleep, while interesting, seemed tangential to the results presented in this manuscript. Please remove.

We have removed some of the discussion of sleep, but have left some to address concerns raised by reviewer 2.